# Hyperuricemia and Hypertension, Coronary Artery Disease, Kidney Disease: From Concept to Practice

**DOI:** 10.3390/ijms21114066

**Published:** 2020-06-06

**Authors:** Mélanie Gaubert, Thomas Bardin, Alain Cohen-Solal, François Diévart, Jean-Pierre Fauvel, Régis Guieu, Stéphane Sadrin, Jean Michel Maixent, Michel Galinier, Franck Paganelli

**Affiliations:** 1Department of Cardiology, Aix-Marseille University, Hôpital Nord, 13015 Marseille, France; melanie.gaubert@ap-hm.fr; 2Department of Rheumatology, Lariboisière Hospital, 75010 Paris, France; thomas.bardin@aphp.fr; 3Department of Cardiology, Lariboisière Hospital, 75010 Paris, France; alain.cohen-solal@inserm.fr; 4Department of Cardiology, Villette Private Hospital, 59240 Dunkerque, France; fdcardio@free.fr; 5Department of Nephrology-Hypertension, Hospices Civils de Lyon, 69002 Lyon, France; jean.pierre.fauvel@chu-lyon.fr; 6Center for CardioVascular and Nutrition Research (C2VN), INSERM, INRA and Aix-Marseille University, 13005 Marseille, France; guieu.regis@orange.fr; 7Unité de recherche Clinique Pierre Deniker (URC C.S. 10587) Centre Hospitalier Henri Laborit, Université de Poitiers, 86021Poitiers, France; Stephane.sadrin@gmail.com (S.S.); jean.michel.maixent@univ-poitiers.fr (J.M.M.); 8I.A.P.S., Equipe Emergeante-Université de Toulon, 83957 Toulon-La Garde, France; 9Department of Cardiology, Rangueil Hospital, 31059 Toulouse, France; galinier.m@chu-toulouse.fr; 10Association pour la recherche et technique médicale, 13015 Marseille, France; 11Department of Cardiology, Nord Hospital, Chemin De Bourelly, 13015 Marseille, France

**Keywords:** serum uric acid, hyperuricemia, cardiovascular disease, renal disease

## Abstract

Since the publication of the Framingham Heart Study, which suggested that uric acid should no longer be associated with coronary heart disease after additional adjustment for cardiovascular disease risk factors, the number of publications challenging this statement has dramatically increased. The aim of this paper was to review and discuss the most recent studies addressing the possible relation between sustained elevated serum uric acid levels and the onset or worsening of cardiovascular and renal diseases. Original studies involving American teenagers clearly showed that serum uric acid levels were directly correlated with systolic and diastolic pressures, which has been confirmed in adult cohorts revealing a 2.21-fold increased risk of hypertension. Several studies involving patients with coronary artery disease support a role for serum uric acid level as a marker and/or predictor for future cardiovascular mortality and long-term adverse events in patients with coronary artery disease. Retrospective analyses have shown an inverse relationship between serum uric acid levels and renal function, and even a mild hyperuricemia has been shown to be associated with chronic kidney disease in patients with type 2 diabetes. Interventional studies, although of small size, showed that uric acid (UA)-lowering therapies induced a reduction of blood pressure in teenagers and a protective effect on renal function. Taken together, these studies support a role for high serum uric acid levels (>6 mg/dL or 60 mg/L) in hypertension-associated morbidities and should bring awareness to physicians with regards to patients with chronic hyperuricemia.

## 1. Background 

Uric acid (UA) is the end-product of purine metabolism, with the last step of xanthine conversion into uric acid being catalyzed by the xanthine oxidase. Elevated levels of serum uric acid (SUA) are considered an important biomarker, and a relationship between SUA has been well established with gout. Gouty arthritis and its mode of action are associated with inflammation, which has been shown to be caused by the deposition of monosodium urate crystals in joints and surrounding tissues both in experimental and clinical studies. A sustained elevated SUA concentration above a threshold value of 6 mg/dL is now accepted [1]. The risk of gout also increases with SUA levels [2]. Overall, other factors responsible for hyperuricemia in humans include ethnicity, age, sex, and genetic and dietary factors [3].

Interests addressing hyperuricemia-induced cardiovascular and renal diseases have dramatically increased over the past decade, suggesting that beyond the risk of gout attack, elevated SUA has been found to be associated with comorbidities including hypertension [3], kidney disease [4], metabolic syndrome [5], type 2 diabetes [3], and heart disease [6]. Results from the National Health and Nutrition Examination Survey (NHANES) cohort including 5707 patients revealed a threefold increase in the prevalence of renal failure, diabetes, and hypertension in the group of patients exhibiting the highest SUA concentrations [7]. 

The relationship between SUA and cardiovascular disease has received a large amount of attention, and the potential role of chronic hyperuricemia as an independent risk factor has become an important issue in defining an appropriate preventive strategy [8]. Although conflicting results have emerged regarding the consequences of hyperuricemia per se on morbidity and mortality after adjustment for traditional cardiovascular risk factors, the most recent studies and meta-analyses mentioned below support the concept of an independent association.

This paper presents an overview of recent studies addressing the impact of chronic hyperuricemia and the expected effect of hypouricemic agents. On this basis, the rationale and principle of a preventive interventional strategy in patients exhibiting elevated SUA levels is discussed. 

## 2. Relation between Serum Acid Uric Concentration and Hypertension-Related Morbidity and Mortality 

### 2.1. A/ Relationship between Serum Uric Acid Levels and Hypertension

This relationship has been evidenced from experimental studies. These studies suggest that hyperuricemia-related hypertension results from a two-phase mechanism, as reviewed and summarized by Feig et al. [9]. First, UA induces a vasoconstriction by activation of the renin–angiotensin system and a reduction of circulating nitric oxide, and hypertension can be reversed by urate-lowering drugs. Second, UA is incorporated into vascular smooth muscle cells, resulting in cellular proliferation. The consequence is a secondary arteriolosclerosis that impairs pressure natriuresis, causing sodium-sensitive hypertension. This process is not reversed by the late reduction of SUA levels. 

From a clinical point of view, this relationship between SUA and hypertension was explored in 125 teenagers for screening of SUA levels and evaluation of hypertension [10] (Figure 1). The study population consisted of children with essential, secondary, or white-coat hypertension. Forty normotensive controls were recruited from the same clinic. Serum uric acid levels were found to directly correlate with systolic (*r* = 0.80, *p* = 0.0002) and diastolic (*r* = 0.66, *p* = 0.0006) blood pressures in controls and in subjects with primary hypertension (but not in subjects with secondary hypertension), independently of renal function. Such a correlation suggested a role for hyperuricemia in the early pathogenesis of primary hypertension. In this study, SUA concentrations > 5.5 mg/dL had a positive predictive value of 82% for primary hypertension. 

This pilot study had some limitations. Although being consistent with the role of hyperuricemia in terms of the risk of hypertension, the pilot study mentioned above involved young people with an age and country-dependent lifestyle known to influence UA metabolism. It is thus important to have data from an adult cohort before discussing the impact on hyperuricemia monitoring. In the context of the 2009–2010 NHANES datasets, 4368 observations out of 10,537 were analyzed after patients less than 20 years old were excluded [11] (Figure 2). Patients with gout or metabolic syndrome, both known to be associated with high SUA levels [12], were also excluded. Hyperuricemia was a significant correlate of hypertension with an adjusted odds ratio of 2.21 (95% CI 2.20 to 3.38). No correlation was found between elevated C-reactive protein and hypertension. These data support the hypothesis that in adults free of metabolic syndrome, elevated SUA level is independently associated with prevalent hypertension.

The meta-analysis for this relationship has been investigated. Using this methodology, the results of this meta-analysis including 25 observational studies with 97,824 patients showed that 11 studies found higher categories of SUA level that were associated with a higher risk of hypertension development, whereas 13 studies showed a continuous and dose-dependent association between SUA level and future hypertension [13]. 

### 2.2. B/ Relationship between Serum Uric Acid Levels and Cardiovascular Risk

This putative relationship between elevated SUA levels and cardiovascular risk has been the subject of debate, especially after the publication of the Framingham Heart study in 1999 [14]. On the basis of a 20-year follow-up of 6763 patients, the authors concluded that after adjustment for age, elevated SUA levels were not associated with an increased risk of an adverse outcome. When examining the data specifically for women after adjustment for age, SUA level was predictive of coronary heart disease and death, a relation no longer observed after adjustment for cardiovascular disease risk factors. The main limitation of the analysis was that data were obtained from only one center in Boston, which included subjects most likely unrepresentative of the whole population, with a lower prevalence of cardiovascular-related deaths.

A meta-analysis based on 11 prospective studies, performed in Europe, the USA, and Asia and involving 584,771 participants, allowed the link between SUA levels and cardiovascular mortality to be revisited [15]. In all the studies taken into account, the data were adjusted for covariates known to impact cardiovascular risk (Figure 3). The results showed that elevated SUA was independently and significantly associated with all-cause mortality (Relative risk (RR) 1.24; 95% CI 1.09–1.42) and cardiovascular mortality (RR 1.37; 95% CI 1.19–1.57). Subgroup analyses showed that the risk of cardiovascular mortality was more pronounced among women (RR 1.35; 95% CI 1.06–1.72). This meta-analysis, including various populations, clearly supports the concept that baseline SUA level is an independent predictor of future cardiovascular mortality. 

Another meta-analysis, based on 33 studies performed worldwide and including 427,917 participants, showed that after adjustment on covariates, hyperuricemia was independently associated with an increased risk of cardiovascular mortality (Hazard Ratio (HR) 1.45; 95% CI 1.18–1.78) and of incident heart failure defined according to Framingham criteria (Hazard Ratio (HR) 1.65; 95% CI 1.41–1.94). Every 1 mg/dL increase in SUA resulted in an increased risk of heart failure of 19% (95% CI 1.17–1.21) [16]. 

The research has progressed and reinforces the relationship between SUA and cardiodiovascular risk. 

### 2.3. C/ Relationship between Uric Acid Levels and Coronary Artery Disease

Again, a constant research interest has focused on this relationship between SUA and coronary artery disease (CAD) and coronary atherosclerosis. A considerable amount of documentation has shown a correlation between SUA levels, recognizing cardiovascular risk factors including age, male sex, hypertension, dyslipidemia, obesity, insulin resistance, hypertension, and diabetes mellitus [17,18]. Because of these relationships, the association between SUA elevations and CAD was considered to be “epiphenomenal” and not causal [19]. However, recent evidence supports the idea that hyperuricemia can be a significant marker and/or an independent risk factor in ischemic heart disease.

Regarding the relationship between SUA levels and coronary atherosclerosis, a number of studies noted significant correlations. Fromonot et al. found that patients with established coronary heart disease have increased SUA levels compared with individuals free of the disease [20]. They also noticed that SUA levels were higher in patients with acute coronary syndrome (ACS) compared with stable CAD patients. 

Deveci et al., using the Gensini score, found a significant association between severity of ischemic heart disease and SUA levels [21]. In contrast, Gur et al. concluded that SUA levels were associated with the presence but not with the severity of CAD [22]. In ACS patients, Duran et al. found a positive association of hyperuricemia with angiographic severity of ischemic heart disease [23]. Recently, Barbieri et al. showed that SUA levels were significantly higher in men than in women, whereas high UA levels were associated with severe CAD only in women [24]. Zhang et al. demonstrated that SUA levels were markedly related to the prevalence of CAD and the development of multivessel disease in premenopausal women, as well as being an independent risk factor for CAD in this specific target group [25]. In these studies regarding SUA and the risk of CAD incidence, hyperuricemia and angiography findings definitions, and variables controlled—such as the use of an antihypertensive drug, a dyslipidemia drug or a diuretic—were not homogenous. Moreover, in these studies physicians did not check if patients were treated with xanthine oxidase inhibitors. Serum uric acid levels after initiating these agents were also not considered.

In line with these previous studies, the hypothesis that elevated SUA levels would be associated with impaired prognostic in CAD patients merits attention. For example, the Multiple Risk Factor Intervention Trial (MRFIT) database demonstrated that hyperuricemia was an independent marker for acute myocardial infarction [11]. Several studies in patients with ST-elevation myocardial infarction (STEMI) treated with primary percutaneous coronary intervention showed that SUA level on admission was a strong independent predictor of major adverse cardiac events [26]. Associated or not with poor coronary blood flow, high SUA level on admission has been independently associated with in-hospital and long-term adverse outcomes [27,28,29]. The impact of hyperuricemia as an independent predictor of cardiovascular mortality has also been confirmed in stable CAD. A study involving 8149 patients with a stable CAD treated with percutaneous coronary intervention showed that SUA levels > 7.5 mg/dL were associated with a 1.6-fold increased risk of mortality (95% CI 1.38–1.86; *p* < 0.001), and that each 1 mg/dL increase in SUA levels was associated with a 31% increase in one-year mortality [30]. 

Although conflicting early data were published regarding the relationship between SUA levels and cardiovascular event risk in the general population, the meta-analyses and individual studies recently published are in favor of a significant relationship between hyperuricemia and ischemic heart disease after adjustment on covariates. The strength of the association found between SUA and morbidity–mortality in groups with established coronary heart disease (CHD) must be highlighted. It supports the notion that a SUA level above a threshold value of 6 mg/dL is an independent predictor of cardiac morbidity and mortality. In light of these studies, SUA level might be an easily measurable biomarker that could be closely monitored for adverse outcomes in the setting of ACS or stable CAD. 

## 3. Relation between Serum Uric Acid Levels and Kidney Disease

Renal physiopathology and SUA represents another part of relationships caused by deleterious action of renal UA and is reviewed later. In patients with kidney disease, the decrease in acid urid urinary excretion is associated with an increase in SUA levels, which may be buffered by gastrointestinal excretory compensation [31]. The stimulation of the renin–angiotensin system induced by hyperuricemia is expected to favor renal vasoconstriction and increase blood pressure, resulting in a vicious circle leading to the progression of renal disease. Several authors have thus addressed the possible impact of asymptomatic hyperuricemia on renal function. 

A retrospective analysis assessed a cohort in the context of work-related health examination [32]. The onset of chronic kidney disease (CKD), defined by an estimated glomerular filtration rate (eGFR) value less than 60 mL/min/1.73 m^2^, was assessed as a primary outcome according to SUA levels > or ≤ 7 mg/dL. Patients with gout were excluded. Multivariate analysis showed an expected relationship between hypertension and the risk of CKD, which increased twofold. More strikingly, SUA levels >7 mg/dL were associated with a fourfold increase (HR 3.99; 95% CI 2.59–6.15) in CKD development, strongly suggesting that asymptomatic hyperuricemia is a predictive factor for new-onset CKD in this population. In a cohort including 2000 subjects without initial renal disease who were followed over 6.5 years, each 1 mg/dL increase in SUA levels was found to be associated with a decrease in 0.19 mL/min/1.73 m^2^ of eGFR [33].

The inverse correlation between SUA levels and renal function is supported by the data of the NHANES cohort collected between 1988 and 2010 from 47,476 subjects [34]. During the most recent period of the survey, the prevalence of gout was less than 5%, whereas that of hyperuricemia reached around 20%. The prevalence of hyperuricemia was found to be dependent on renal function, being 11%–13% in subjects with eGFR ≥ 90 mL/min/1.73 m^2^, but reaching 64%–78% in subjects with eGFR less than 30 mL/min/1.73 m^2^ (Figure 4).

Given the relationship of some commonly prescribed antihypertensive agents with incident gout and increased SUA levels [5], the strong correlation between hyperuricemia and renal failure grade should be taken into account by physicians when choosing therapy for patients with CKD. 

In conclusion, SUA during kidney pathophysiology and therapy should retain continuous attention.

## 4. What Is the Expected Impact of Lowering Serum Uric Acid Levels on the Risks Associated with Chronic Hyperuricemia? 

Several lines of clinical evidence during hyperuricemia, hypouricemia, and therapies should help us to answer this important question. This chapter focuses on the clinical part of this relationship.

In order to validate the implication of hyperuricemia in cases of hypertension and related comorbidities, several authors have assessed the impact of lowering SUA levels. Most of the studies assessed the effect of xanthine oxidase (XO) inhibitors, a class of urate-lowering drugs (ULD) approved in the management of gout. Several studies demonstrated that the decrease in SUA induced by XO inhibitors had a positive impact on the progression or the incidence of the corresponding diseases.

In a short-term crossover study involving adolescents with newly diagnosed hypertension, treatment with 200 mg allopurinol twice a day resulted in a reduction of SUA levels, which was associated with a suppression of hypertension in 20 out of 30 patients treated [35]. The increase in diastolic or systolic blood pressure could be significantly prevented by lowering SUA levels by the same XO inhibitor, which supports the concept that an early intervention aimed to prevent hyperuricemia may be beneficial regarding hypertension and its comorbidities.

One of the most demonstrative studies on the impact of hypouricemics on hypertension-related diseases was published by Goicoechea et al [31]. In this prospective trial, 113 patients with estimated eGFR < 60 mL/min/1.73 m^2^ were randomized to receive allopurinol 100 mg/day or to continue their usual therapy. Over a mean follow-up of 23 months, SUA levels were unchanged in the control group, whereas they were reduced to a normal value of 6 mg/dL in the XO inhibitor group. The hypouricemic treatment was associated with a significantly decreased risk of cardiovascular events and hospitalization. Regarding the impact on renal function, the changes of SUA levels and eGFR were significantly and inversely correlated (*r* = −0.375, *p* = 0.001) (Figure 5). The eGFR value was not significantly changed in the allopurinol group, whereas it was worsened in the control group, suggesting that lowering SUA by using ULD is expected to slow the progression of renal disease. Such an impact on the renal function of lowering SUA has also been observed in the post-hoc analysis of phase III studies assessing febuxostat (80-120 mg/day), another XO inhibitor, in patients with gout over a four-year period [36]. In this study, the greatest decreases in SUA levels were associated with less renal function decline defined by MDRD-eGFR (*p* < 0.001). It was estimated that for every 1 mg/dL of chronic reduction of SUA level, there would be a preservation of 1.15 mL/min/1.73 m² of eGFR. The positive impact of XO inhibitor-based intervention on renal function is also supported by a recent prospective randomized pilot study assessing the impact of febuxostat versus placebo in 60 hypertensive and hyperuricemic patients. The dose of febuxostat was adjusted to maintain the SUA level at < 6.0 mg/Dl [37]. Results over six months showed a reduction of plasma renin activity and aldosterone concentrations, and a significant increase in eGFR (+5.5%, *p* = 0.001) observed in the placebo group. This study suggests that the decrease in SUA induced by febuxostat may allow a suppression of the renin–angiotensin–aldosterone system and an improvement in renal function in hypertensive patients. 

Although strongly suggesting that the benefits of allopurinol and febuxostat on hypertension and its comorbidities were mediated by a reduction of SUA levels, it can be postulated that the effect resulted from another action of the XO inhibitors. The confirmation that the protective effects observed were mediated by SUA reduction is provided by a randomized double-blind study involving 60 prehypertensive obese adolescents, comparing the effect of allopurinol with probenecid [38], and comparing both with a placebo. Over two months, both urate-lowering therapies were associated with a significant decrease in blood pressure. These results provide indirect evidence that UA increases blood pressure in adolescents, and that the reversion of this effect is mediated by SUA reduction (Table 1).

The results of the studies mentioned above are reinforced by those based on the U.K. Clinical Practice Research Database comparing cardiac-related outcomes of patients with hypertension receiving or not receiving allopurinol [39]. This intervention-based analysis in a real-life setting showed that allopurinol use was associated with a significantly lower risk of both stroke (HR 0.50; 95% CI 0.32–0.80) and cardiac events (HR 0.61; 95% CI 0.43–0.87), and supported the hypothesis that long-term pharmacological inhibition of XO may decrease the risk of coronary and cerebrovascular disease. 

If the previous studies mentioned above highlight the positive impact of XO inhibitors on hypertension and renal function, a recent noninferiority trial demonstrated the importance of XO inhibitor choice. A total of 6190 patients with gout and cardiovascular disease were randomized to receive febuxostat (40 to 80 mg daily) or allopurinol (200 to 600 mg) during a median follow-up of 32 months [40]. The incidence of the primary composite endpoint was similar in febuxostat (10.8%) and allopurinol (10.4%) (*p* = 0.002 for noninferiority). However, the febuxostat group, with a greater urate lowering, had significantly higher rates of death from all causes (7.8% vs. 6.4%, *p* = 0.04) and from cardiovascular causes (4.3% vs. 3.2%, *p* = 0.03). Such surprising results need further safety analyses to evaluate the unexpected mortality findings. It must be noted that a high rate of discontinuation (45%) was encountered in this trial.

Taken together, these studies highlight the importance of lowering SUA in hypertensive hyperuricemic patients, especially if done at an early stage to prevent hypertension-associated comorbidities.

## 5. Is There a Pathogenetic Role for Serum Uric Acid in Cardiovascular Disease?

Hyperuricemia has a strong genetic basis, and the heritability of serum urate concentrations is estimated at 40%–70% [41], which justifies the search for its genetic determinants [42]. The main confounding factor might be that SUA has an evident heritable comportement with an overall heritability of 40%-60% [43]. The genetic and environmental contribution to hyperuricemia are therefore likely to be underestimates, because of the potential biochemical measurement errors engendered by wide intervals between study examinations. The present review of observational and/or randomized studies revealed that there is a strong relationship between SUA levels and cardiovascular risk and mortality, with a threshold value of 6 mg/dL. Observational studies might be affected by residual confounding factors and undetected bias. Confounding factors could be attenuated with pooled adjust association. For example, ischemic heart disease without adjustment for cardiovascular risk showed a relative risk of 1.34 versus 1.09 after adjustment [43]. Randomized controlled trials eliminate bias and confounding factors, but ethical concerns limit their application. The role of SUA as an independent risk factor for cardiovascular disease therefore remains unclear. As an alternative to randomized controlled trials, Mendelian randomization (MR) design has been developed for exploring the causal effect of a biomarker, SUA in particular. MR studies can provide a cost-effective analogy to randomized controlled trials using genetic variants as proxies to test the causality of this association. Genetic markers could be exploited for testing the hypothesis that hyperuricemia is causally implicated in cardiovascular disease. Around 30 gene variants have been identified, explaining about 7% of the variation in SUA. Kei et al. recently reviewed the current literature in MR studies regarding SUA levels and cardiovascular disease [44]. Genetic evidence suggests a modest causal effect of SUA levels on cardiovascular disease but confounding polymorphism linkage, sex-specific effects must be interacted. Recently, Li et al. [45] carried out an umbrella review of meta-analyses of MR studies on the association between SUA and cardiovascular outcomes. Regarding CAD, in contrast with the meta-analyses, MR studies reported disparate results. This could be explained by the fact that most of these trials were unable to detect modest effects.

## 6. Perspectives: Lower and Stable Serum Uric Acid Levels? 

Whether UA truly represents an independent risk factor for the development of cardiovascular disease (CVD) is still a controversial matter, despite the Mendelian randomization (MR) or large meta-analyses of prospective cohort studies. The debate of hyperuricemia and cardiovascular risk is open now (for a recent review) [46]. Modern lifestyles and the Western diet—rich in uric acid-raising components such as red meat or sugar, especially fructose—are accompanied by a dramatic rise in SUA levels.

Is elevated UA a risk marker or risk factor for CAD? At present, no definitive conclusion can be drawn as to whether or not UA is an independent predictor of CAD. Even meta-analyses are contradictory. The answer is far from simple and comes from both experimental and clinical studies. Experimental research suggests that UA exerts opposite effects on oxygen scavenging, according to whether its effects occur extracellularly or intracellularly. Regarding experimental studies, circulating UA protects vascular endothelial cells from oxidative stress, and when it penetrates the endothelium, the effects of UA are completely different and it becomes a strong pro-oxidant [9]. Reactive oxygen species (ROS) including O_2_− and H_2_O_2_ play a major role in cardiac dysfunction and reduce nitric oxide bioavailability. Furthermore, hypouricemia such as XO inhibitor treatment has been shown to improve endothelial dysfunction. These data therefore suggest that UA may play a role in the inflammation process that occurs during ischemia. Regarding clinical studies, to define with certainty a risk factor, it has to be demonstrated that reducing it improves the prognosis. Prognostic data, unfortunately, are not available. Previous studies have observed that allopurinol abolishes vascular oxidative stress and improves endothelial-dependent vasodilation and pulsed wave analysis, as well as abolishing vascular oxidative stress, giving rise to the speculation that it may lead to a possible mortality reduction in patients with CAD. The number of patients was small (*n* = 66) with follow-up limited (between 6 weeks and 9 months), but it has been speculated that urate-lowering therapy (ULT) may lead to a possible mortality reduction in patients with CAD. Encouragingly, a prospective cohort study (*n* = 7135) demonstrated that high-dose allopurinol treatment is associated with a lower risk of cardiovascular events and mortality. Unfortunately, other studies have reached the opposite conclusion. 

Although previous studies have not yet provided direct evidence that ULT reduces the risk of CHD mortality in hyperuricemic patients, the studies discussed above provided some positive data. Possible explanations for diverging results could be sex-specific effects, an interaction with kidney function, or the possibility that UA might turn pro-oxidant under certain conditions (e.g., CAD). Whether lowering UA in ischemic heart disease might be clinically beneficial remains to be elucidated in large prospective clinical trials.

Increased variability in cardiovascular parameters, including blood pressure, body mass index (BMI), amd glucose and cholesterol levels, is associated with an increased risk of future cardiovascular events. Furthermore, a surge of UA in the blood is known to increase the crystallization rate of urate by stimulating an immune reaction and inflammatory response. It has therefore been speculated that large fluctuations in UA levels can be deleterious in patients with CAD. The underlying mechanism might be the magnitude of activation of the ROS system, which may contribute to disease progression. More studies linking higher SUA variability with increased higher future risk events considering medication and underlying comorbidities are needed. This highlights the importance of achieving stable SUA levels and avoiding large fluctuations.

Even if current available data are not definitive, several actions of acid uric suggest its involvement in the development and progression of CAD. Hence, further research is necessary. The increasing evidence that SUA plays a key role in cardiovascular disease and both existing and emerging medicines to modify its concentration warrants in-depth investigations to further elucidate answers in the long-lasting debate about SUA and CVD. Currently, the ALL-HEART study, investigating the benefit of allopurinol as an addition to secondary prevention therapy, is recruiting. This study is an academic clinical trial that aims to address whether allopurinol added to usual therapy improves cardiovascular outcomes in patients with ischemic heart disease.

## 7. Perspectives and ALL-HEART Study 

Although numerous publications have documented the link between UA and blood pressure, the issue of hyperuricemia is usually restricted to gout arthritis, the management of which is too often focused on the treatment of gout flares. The consequence of this misconception is that the long-term impact of chronic hyperuricemia on monosodium urate crystal deposition and on the development of renal and/or cardiovascular diseases is not perceived. Having been the subject of debate for several years, the relationship between sustained SUA above the threshold value of 6 mg/dL and renal dysfunction or cardiovascular events including deaths and stroke is now supported by several robust studies including updated registers such as the NHANES and meta-analyses, which allows for analyses and interpretations based on large cohorts of representative patients. Concurrently, it has become clear that lowering SUA by XO inhibitors may prevent these complications. Targeted lowering and systematic monitoring of SUA is now recommended in gouty patients, and the implementation of a similar approach in asymptomatic hyperuricemic patients at risk of hypertension-associated comorbidities should be tested. Physicians should be aware of the robust demonstration that patients with a SUA above 6 mg/dL are at an increased risk of developing hypertension, as well as heart or renal diseases. Lifestyle modifications are already proven to decrease uricemia and cardiovascular risk. ULD is licensed for the prevention of gout but is not currently indicated for treating asymptomatic hyperuricaemia. Allopurinol has several beneficial effects in cardiovascular disease and is usually given in doses of 100–900 mg daily. Against a background of many years of supportive evidence for the potential benefits (morbidity, quality of life, tolerability, safety) of allopurinol in cardiovascular disease, the ALL-HEART study should be a key outcome study that should answer the question of whether adding allopurinol therapy to usual care in patients with ischemic heart disease aged over 60 years improves major cardiovascular outcomes. Furthermore, the ALL-HEART study (multicenter, controlled, prospective, randomized) open-label blinded end point (PROBE) trial of allopurinol 600 mg daily versus no treatment also provides robust data on any improvements in quality of life, as well as an analysis of the health economics of the use of allopurinol. It is necessary to demonstrate that pharmacological reduction of SUA levels improves outcomes in patients with cardiovascular diseases, especially in those with hyperuricemia without gout. Patient recruitment for the ALL-HEART study started in 2014, and results are expected near 2021. 

## 8. Conclusions

In sum, in the light of the present review study, and in accordance with the task force for the management of arterial hypertension of the European Society of Cardiology (ESC) and the European Society of Hypertension (ESH), we suggest that SUA level—a cheap, routinely assessed, and easily measurable biomarker—represents a potentially treatable target, and that lowering SUA in CAD might be clinically beneficial. 

## Figures and Tables

**Figure 1 ijms-21-04066-f001:**
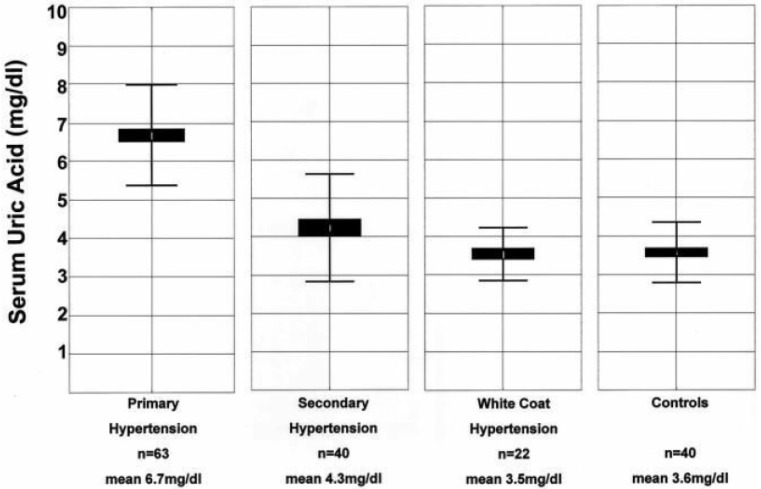
Box-and-whisker plot of serum uric acid levels in children with hypertension and normal blood pressure. The mean and SD for serum uric acid (SUA) levels for primary, secondary, and white-coat hypertension, and controls are shown [10].

**Figure 2 ijms-21-04066-f002:**
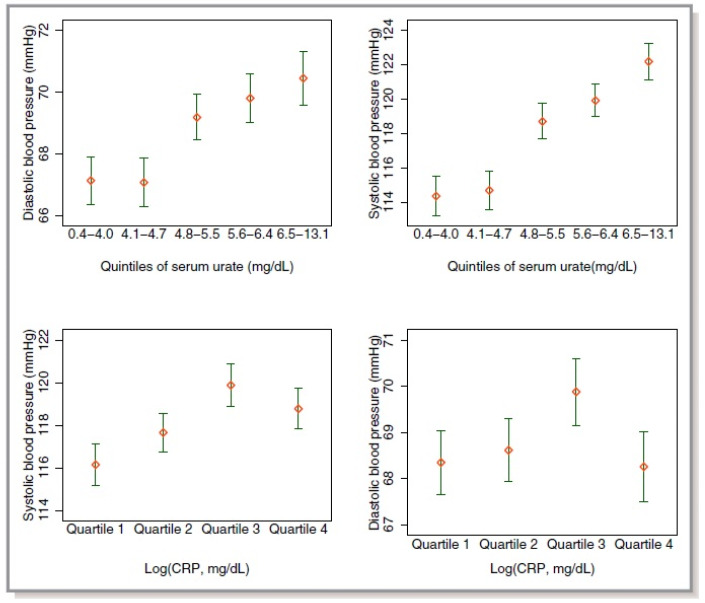
Bivariate associations between log (C-reactive protein), serum urate, and blood pressure. Mean and 95% CIs were calculated using survey weights [11].

**Figure 3 ijms-21-04066-f003:**
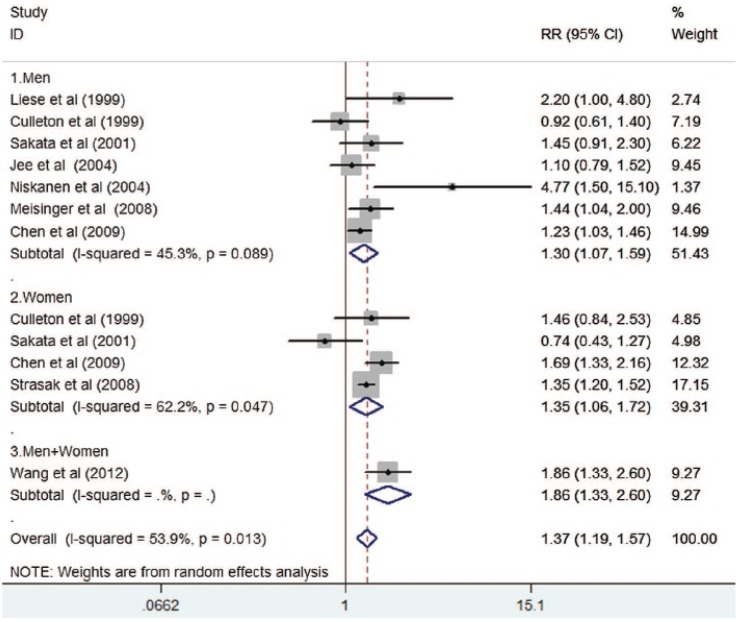
RR and 95% CI from the eligible studies of elevated serum uric acid level and cardiovascular mortality comparing the highest serum uric acid to the lowest category group in a random effect model [15].

**Figure 4 ijms-21-04066-f004:**
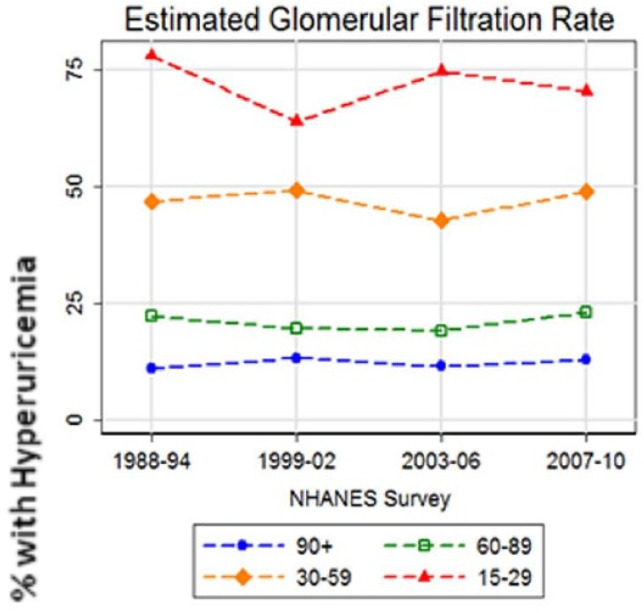
Prevalence of hyperuricemia stratified by estimated glomerular filtration rate [34].

**Figure 5 ijms-21-04066-f005:**
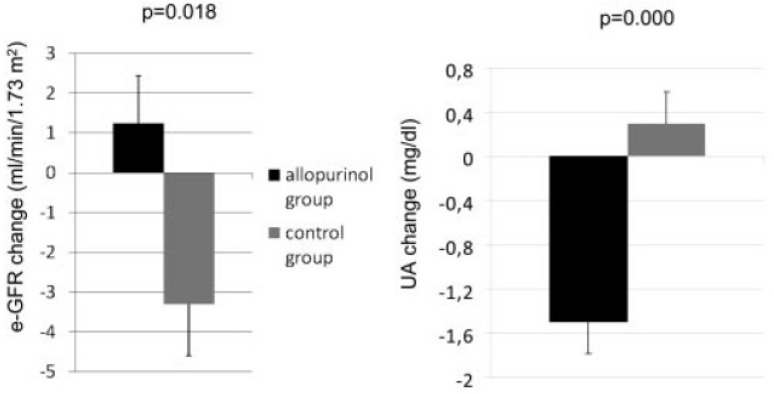
Change in uric acid levels and estimated glomerular filtration rate (eGFR) at the end of study [31]. Values are expressed as mean ± SEM.

**Table 1 ijms-21-04066-t001:** End points at the conclusion of the two-month treatment phase [38].

* End Point (units)	Placebo	*P* ^†^	Allopurinol	*P* ^†^	*P* Allopurinol vs. Placebo
Serum uric acid, mg/dL	6.3 (5.8 to 6.8)		4.1 (3.4 to 4.7)		
(Change from baseline)	−0.3 (−0.1 to −0.7)	0.048	−2.8 (−2.1 to −3.6)	0.0003	0.0005
^‡^ (Adjusted change)	−0.4 (−0.2 to −0.7)	0.046	−2.8 (−2.2 to −3.4)	0.0003	0.0008
SBP, mm Hg	128.1 (124.8 to 131.4)		118.1 (113.2 to 120.1)		
(Change from baseline)	+1.7 (0.9 to 3.8)	0.037	−10.1 (−8.2 to −13.1)	0.0002	0.0001
(Adjusted change)	+1.0 (0.4 to 3.1)	0.071	−10.3 (−8.3 to −13.5)	0.0002	0.0003
DBP, mm Hg	76.0 (73.6 to 78.4)		66.2 (62.7 to 70.7)		
(Change from baseline)	+1.6 (0.2 to 3.5)	0.033	−8.0 (−5.8 to −10.1)	0.0006	0.0002
(Adjusted change)	+1.3 (0.2 to 3.0)	0.057	−8.0 (−5.4 to −10.7)	0.0006	0.0007
24-h SBP, mm Hg	120.0 (114.9to125.1)		113.5 (110.3 to 117.3)		
(Change from baseline)	+1.9 (−0.4 to 2.4)	0.310	−9.2 (−6.7 to −11.3)	0.0004	0.0008
(Adjusted change)	+1.1 (−0.8 to 2.1)	0.382	−9.5 (−6.8 to −11.9)	0.0003	0.0008
24-h DBP, mm Hg	68.7 (66.4 to 71.1)	0.262	62.4 (60.4 to 64.5)		
(Change from baseline)	+1.3 (0.2 to 3.5)	0.262	−6.1 (−4.6 to −9.0)	0.0009	0.0011
(Adjusted change)	+0.9 (0.0 to 3.3)	0.322	−6.4 (−4.7 to −9.5)	0.0007	0.0014
Weight, kg	99.8 (87.6 to 109.2)		98.1 (81.2 to 110.4)		
(Change from baseline)	+2.1 (1.7 to 2.8)	0.008	−0.9 (−1.5 to −0.6)		
(Adjusted change)	N/A		N/A	0.42	0.039
BMI, kg/m^2^	36.6 (32.3 to 39.9)		36.1 (33.5–41.0)		
(Change from baseline)	+0.9 (0 to +1.2)	0.060	−0.2 (−0.5 to +0.7)	0.61	0.041
(Adjusted change)	N/A		N/A		
SVRi^**^ (dyne s/cm^5^)/m^2^	2271 (2017 to 2527)		1761 (1175 to 1946)		
(Change from baseline)	+8.6% (−0.2% to 12.2%)	0.145	−24.5% (−49.6% to −16.5%)	0.0006	0.0004
(Adjusted change)	+2.3% (−0.8% to 5.1%)	0.344	−25.1% (−49.8% to −16.1%)	0.0006	0.0003

BMI, body mass index; DBP, diastolic blood pressure; SBP, systolic blood pressure; SVRi, systemic vascular resistance index; N/A, not applicable. Parentheses delineate the 95% CI. * End points are the mean value the end of the treatment. † *p*-values are two- tailed *t*-test comparison of the mean change from pretreatment value. ‡ Values were adjusted for weight and BMI.

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
