# Peer review of "Hyperuricemia and Hypertension, Coronary Artery Disease, Kidney Disease: From Concept to Practice"

_ijms, 2020, doi:10.3390/ijms21114066_

Round 1

Reviewer 1 Report

In this review Gaubert, et al provide an overview of the data relating to uric acid concentrations and cardiovascular risk. In general the critical information and key points are discussed but the structure and flow of this review makes it difficult to read. For example, the background section begins by discussing the current trend in publications on uric acid and cardiovascular disease and an overview of gout before discussing the basic biology/physiology if uric acid production as a by-product of xanthine conversion. Similarly, the section that explores the relationship between uric acid concentration and cardiovascular risk should come before the section that of the relationship to hypertension.

There are stylistic contrasts from paragraph to paragraph that give the sense of a number of blocks of information put together rather than a cohesive narrative on the current knowledge and clinical practice.

As mentioned, the key evidence is presented within this text but the manner in which it is included reads more like a regurgitation of previous studies with little narrative or contribution from the authors of this review to stitch it together into a flowing review article.

One very important aspect is not discussed here- the authors finish by concluding that serum uric acid should be screened in CAD patients but have not discussed that the 2018 ERC guidelines for arterial hypertension management make the same suggestions in hypertensive patients.

Numerous minor revisions of the use of English are required, some abbreviations are not defined in the text and are inconsistently used throughout the text.

This is a review on a topical and important subject area that is likely to be of interest to the field. The current version has the foundations of a strong article but needs extensive restructuring for the information to flow in a cohesive and readable manner.

Author Response

Referee 1

In this review Gaubert, et al provide an overview of the data relating to uric acid concentrations and cardiovascular risk. In general the critical information and key points are discussed but the structure and flow of this review makes it difficult to read. For example, the background section begins by discussing the current trend in publications on uric acid and cardiovascular disease and an overview of gout before discussing the basic biology/physiology if uric acid production as a by-product of xanthine conversion. Similarly, the section that explores the relationship between uric acid concentration and cardiovascular risk should come before the section that of the relationship to hypertension. There are stylistic contrasts from paragraph to paragraph that give the sense of a number of blocks of information put together rather than a cohesive narrative on the current knowledge and clinical practice.As mentioned, the key evidence is presented within this text but the manner in which it is included reads more like a regurgitation of previous studies with little narrative or contribution from the authors of this review to stitch it together into a flowing review article.

Answer : We agreed with the reviewer’s comments that our review should be revised for a convieninent reading. The background and the following chapters has been reorganized with introduction from the basic rationale with uric acid to clinical research and practice to have a more flowing review article. This is underlined by blue changes in the main text.

One very important aspect is not discussed here- the authors finish by concluding that serum uric acid should be screened in CAD patients but have not discussed that the 2018 ERC guidelines for arterial hypertension management make the same suggestions in hypertensive patients.

Answer : This important has been added in our short and new focused conclusion.

Numerous minor revisions of the use of English are required, some abbreviations are not defined in the text and are inconsistently used throughout the text.

Answer ; these mistakes have been corrected and improved by a professional native english people (All the changes are shown in yellow).

This is a review on a topical and important subject area that is likely to be of interest to the field. The current version has the foundations of a strong article but needs extensive restructuring for the information to flow in a cohesive and readable manner.

Answer : Thanks for your recommendation, we agreed that our manuscript could be improved. We improved it for a flowing reading by many ways (see the revised manuscript with in red and yellow our modifications).

Reviewer 2 Report

Dear sir, thank you to select me to review  manuscript: Hyperuricemia and hypertension, coronary artery disease, kidney disease: from concept to practice. Review is concise and well written. Only some recommendation to improve quality of this artice are recommended: 

1) Line 49: add sentence Levels of SUA also depend on etnicity, age and sex; Petrikova J et al. Int J Environ Res Publ Health. 2018 Jul 4;15(7):1412.doi: 10.3390/ijerph15071412.

2) Line 168: add  chapter about role of genetics in assocation betwenn SUA and cardiovascular risk (Kei A et al. Int J Clin Pract. 2018 Jan;72(1). doi: 10.1111/ijcp.13048; Kleber ME et al. Am J Soc Nephrol. 2015 Nov;26(11):2831-8. doi: 10.1681/ASN.2014070660; Palmer TM et al. 2013 Jul 18;347:f4262.doi: 10.1136/bmj.f4262 and other articles).

3) Chapter: Conclusions should be shortened.

4) Abbreviations: add SUA, this is not a standard abbreviation. 

Author Response

Hyperuricemia and hypertension, coronary artery disease, kidney disease: from concept to practice. Review is concise and well written. Only some recommendation to improve quality of this artice are recommended: 

Answer : Thanks for your recommendation, we agreed that our manuscript could be improved. We improved it for a flowing reading by many ways (see the revised manuscript with in blue and green our modifications).

1) Line 49: add sentence Levels of SUA also depend on etnicity, age and sex; Petrikova J et al. Int J Environ Res Publ Health. 2018 Jul 4;15(7):1412.doi: 10.3390/ijerph15071412.

2) Line 168: add  chapter about role of genetics in assocation betwenn SUA and cardiovascular risk (Kei A et al. Int J Clin Pract. 2018 Jan;72(1). doi: 10.1111/ijcp.13048; Kleber ME et al. Am J Soc Nephrol. 2015 Nov;26(11):2831-8. doi: 10.1681/ASN.2014070660; Palmer TM et al. 2013 Jul 18;347:f4262.doi: 10.1136/bmj.f4262 and other articles).

Answer : The sentence in 1) has been corrected and line 168 in the main text, this pertinent reference was cited and added as reference 3 in the references list.

2) a new chapter on genetic aspects including 5 new references (listinf as references 41-45) has been done according to these pertienent comments.

3) Chapter: Conclusions should be shortened.

Answer ; A clear conclusion in two points (including the important and recent clinical guideline) has been shortened. A chapter on the prespective for the ALL_HEART study has been created before the conclusion.

4) Abbreviations: add SUA, this is not a standard abbreviation

Answer : line 864, we add « SUA : serum uric acid »

Round 2

Reviewer 1 Report

The alterations made to this paper have improved the flow and readability of the content. There are still some areas where this could be improved but overall it is an interesting and insightful review.